# Hsa-mir-135a Shows Potential as A Putative Diagnostic Biomarker in Saliva and Plasma for Endometriosis

**DOI:** 10.3390/biom12081144

**Published:** 2022-08-19

**Authors:** Alexandra Perricos, Katharina Proestling, Heinrich Husslein, Lorenz Kuessel, Quanah J. Hudson, René Wenzl, Iveta Yotova

**Affiliations:** Department of Obstetrics and Gynecology, Medical University of Vienna, 1090 Vienna, Austria

**Keywords:** miRNA, hsa-mir-135a, noninvasive diagnostic biomarker, endometriosis

## Abstract

Endometriosis is a chronic disease characterized by the implantation and proliferation of endometrial tissue outside of the uterine cavity. The nonspecific nature of the symptoms and the lack of sensitive, noninvasive diagnostic methods often lead to a significant delay in diagnosis, highlighting the need for diagnostic biomarkers. The correlation of circulating miRNAs with altered inflammatory signals seen in patients with endometriosis has raised the possibility that miRNAs can serve as biomarkers for the disease. In our study, we analyzed miRNA expression in saliva of women with and without endometriosis using a FireFly custom multiplex circulating miRNA assay. This focused panel included 28 human miRNAs, 25 of which have been previously found to be differentially expressed either in plasma, serum, and/or blood of women with endometriosis, compared to controls. We found that hsa-mir-135a was expressed significantly higher in the saliva of women with endometriosis, independent of disease stage and menstrual cycle phase. We confirmed that hsa-mir-135a also showed significantly elevated expression in the plasma of endometriosis patients. This indicates that hsa-mir-135a is a putative noninvasive biomarker of endometriosis in both saliva and plasma, but further validation studies are required to assess its clinical value as a biomarker.

## 1. Introduction

Endometriosis is an estrogen-driven chronic disease that affects up to 10% of women of reproductive age [1]. It is characterized by the implantation and proliferation of endometrial tissue outside of the uterine cavity, accompanied by dysregulation of steroid hormone production and response in eutopic endometrium and ectopic lesions [2]. The symptoms are heterogeneous in type and severity and include dysmenorrhea, dyspareunia, chronic pelvic pain, and infertility [3]. While highly dependent on the experience of the examining physician, deep infiltrating endometriosis and ovarian endometriotic cysts may be diagnosed by ultrasound or magnetic resonance imaging, but these methods fail to diagnose superficial peritoneal ectopic lesions [4]. Thus, currently, the gold standard for diagnosis of this disease remains visualization through laparoscopic surgery and histological confirmation. The nonspecific nature of the symptoms and the lack of sensitive, noninvasive diagnostic alternatives often lead to a significant delay in diagnosis and onset of treatment [1].

During the last decade, the effort for developing new noninvasive diagnostic tools for endometriosis has dramatically increased. MicroRNAs (miRNAs) have emerged as promising candidates based on their function as important epigenetic regulators of genes and signaling pathways associated with the development and survival of endometriotic lesion [5,6,7,8]. Within cells, these small (22 nucleotides (nt) in length), single-stranded molecules act at the posttranscriptional level to either repress transcription or degrade their target messenger RNA (mRNA) [9]. MiRNAs have also been identified circulating in various body fluids in a cell-free form [10]. Most of the circulating miRNAs are included in lipid or lipoprotein complexes, such as apoptotic bodies, microvesicles, or exosomes, and are therefore highly stable [10]. The correlation of circulating miRNAs with the altered inflammatory signals seen in patients with endometriosis has raised the possibility that miRNAs can serve as specific diagnostic biomarkers for the disease [11,12]. For example, Nematian, S. E. et al. [11] showed that the circulating miRNAs 125b-5p and Let-7b-5p function as regulators of macrophage inflammatory cytokine production in women with endometriosis compared to controls. To date, around 15 studies, including our own, have evaluated the potential of circulating miRNAs as diagnostic markers for endometriosis, with several single or panels of miRNAs showing promising diagnostic properties [13,14,15]. While several studies have evaluated miRNA patterns in blood associated with endometriosis, the miRNA signature in saliva is still understudied. The diagnostic power of salivary miRNAs has been investigated in the context of cancer and other local and systemic disorders [16,17,18], indicating the potential value of this approach.

Saliva presents advantages over plasma/serum as a source for biomarkers, with collection being simple, safe, noninvasive, cost-effective, and not requiring medical personnel. Therefore, in this study, we used a focused panel of miRNAs identified as potential endometriosis biomarkers in other tissues to assess their expression in the saliva of endometriosis patients and controls.

## 2. Materials and Methods

### 2.1. Study Population

Women between 18 and 50 years of age undergoing laparoscopic surgery due to suspected endometriosis, pelvic pain of unknown origin, adnexal cysts, infertility workup, or uterine fibroids were included in this study. Based on laproscopic visual diagnosis and histological confirmation, the women were assigned as either control or endometriosis patients.

Patients were excluded from the study if they had undergone hormonal therapy during the three months before the surgery, had any history of malignant disease, had a current infection or inflammatory condition, or had a systemic autoimmune disorder. All patients were asked to fill in a detailed questionnaire in order to evaluate pain symptoms potentially associated with endometriosis (assessed using the visual analogue scale (VAS), 0 = no pain; 10 = worst possible pain) as well as a detailed patient history sheet, resulting in a well-characterized patient cohort. The presence or absence of endometriosis was confirmed laparoscopically and by histopathological analysis. The different stages of endometriosis disease were classified according to the revised American Society of Reproductive Medicine (rASRM) [19]. Patients who did not show any endometriotic lesions by laparoscopic evaluation were included in the control group. The menstrual cycle phase on the day of the surgery was evaluated either by histologic analysis of an endometrial biopsy for women undergoing diagnostic curettage or based on information provided by the patient about the duration of the menstrual cycle and the day after last menstruation where no diagnostic curettage was available. From the 40 women who participated in the study, conducted from August 2019 through March 2020, 34 women fulfilled all criteria and were included in the study.

### 2.2. Institutional Ethics Committee Approval

The study was approved by the institutional ethics committee of the Medical University of Vienna (EK1398/2019). All patients gave their written informed consent prior to inclusion in this study.

### 2.3. Sample Collection and Preparation for Analysis

Saliva and plasma samples were collected from participating patients preoperatively in a fasting state on the day of surgery. To avoid circadian-rhythm-associated differences in the levels of miRNAs expression, all samples were collected in the morning over a maximum time frame of 6 h. The saliva was collected using the “passive drool” method as recommended by the World Endometriosis Research Foundation for general biomarker studies [20]. All collected samples were centrifuged at 1000× *g* for 10 min at room temperature, the supernatants were removed, and the samples were aliquoted and frozen at −80 °C until further analysis. Note that two hemolytic plasma samples were excluded from the analysis (one control and one endometriosis sample).

### 2.4. Focused miRNA Panel and FireFly microRNA Profiling Assay

The focused panel of miRNAs in this study targets 28 human miRNAs, 25 of which have been previously found to be differentially expressed either in plasma and/or serum or blood of women with endometriosis, compared to controls (Table 1). Fifteen of these miRNAs have shown promising diagnostic properties for the disease, either as a single miRNA or in a panel with other differentially expressed miRNAs (Table 1). Hsa-miR-30e was included as putative normalization control together with hsa-mir-16-5p, hsa-mir-103a-3p, which in some studies were differentially expressed but in others were used as internal normalizers (Table 1). As hsa-miR-584 has already been reported to be expressed in saliva, this miRNA was included in the panel as a positive control [21]. The cell-mir-39-3p miRNA served as an external spike in the control.

We used a FireFly custom multiplex circulating miRNA assay (Abcam, Cambridge, MA, USA) [33] for the analysis of the levels of expression of our focused panel of miRNAs in saliva of women with and without endometriosis. The advantage of this method over widely used qPCR-based microarrays or deep sequencing approaches is the possibility of multiplex determination of the targets using small amounts of starting patient material, as well as a lower price. The starting material for this analysis was 30 µL of saliva. Samples were run in duplicates at the Abcam Service Laboratory; the raw data subtracted for background signal and then further analyzed in our laboratory.

### 2.5. Expression Arrays Data Normalization (Preprocessing)

Prior to data normalization, the mean expression level for each of the miRNAs was calculated from technical duplicates for each sample. First, we used several methods to determine if our dataset included miRNAs that could be used as internal normalization controls. miRNAs internal normalization controls should be stably expressed, normally distributed across all samples, regardless of study group, and should show low variability. The FireFly Workbench software geNorm-like algorithm based on these principles found no suitable normalizers. Further analysis of the variability and normal distribution of miRNA expression conducted using SPSS software for Windows (Version 17.0; IBM Corp., Armonk, NY, USA) [34] confirmed that our dataset contained no suitable miRNA internal normalization controls (Appendix A). Therefore, the data were normalized to the mean of the cel-miR-39-3p spike in control. To validate the results, we reanalyzed the data using alternative normalization methods. In the first alternative approach, we normalized to the arithmetic mean of the expression levels of the 28 miRNAs plus the spike in miRNA. In the second approach, we normalized to the geometric mean of these miRNAs. The standard geometric mean cannot be calculated from a dataset that contains zero values [35]. Therefore, we calculated a nonstandard geometric mean removing the zero values as previously described [36].

### 2.6. Statistics

All statistical tests were performed using SPSS statistics software for Windows (Version 17.0; IBM Corp., Armonk, NY, USA). Nonparametric Mann–Whitney U-tests and the Bonferroni–Holm algorithm for multiple testing were applied to compare the groups. Receiver operating characteristics (ROC) analysis was used to examine the diagnostic value of the candidate miRNAs. Differences with an adjusted *p*-value (adj. *p*) < 0.05 were considered significant.

## 3. Results

### 3.1. Patient Characteristics

Patients were subject to laparoscopic examination and histological confirmation for endometriosis. Of the initial 40 patients recruited from the study, 34 were retained following the exclusion criteria described in the methods. From these, 17 had endometriosis and 17 showed no evidence of endometriosis; therefore, the latter represented the control group. The patient characteristics are shown in Table 2.

### 3.2. Differentially Expressed miRNAs in Saliva of Women with Endometriosis

All tested miRNAs were expressed in saliva of our patient collective (Appendix A). Hsa-mir-16-5p and hsa-mir-191-5p were detected in all samples, while hsa-mir-145-3p was most rarely expressed being detected in only 21% of the samples. We further analyzed the differences in the levels of expression of miRNAs between patients and controls and found hsa-mir-135a to be expressed significantly higher in saliva of women with endometriosis, compared to controls (adj. *p* = 0.023, Figure 1A). Reanalyzing the data with alternative normalization methods produced similar results (normalization to arithmetic mean, adj. *p* = 0.015, Appendix A, normalization to nonstandard geometric mean, adj. *p* = 0.054, Appendix A). This difference was independent of the stage of the disease (Figure 1B) and menstrual cycle phase (Figure 1C). In addition, hsa-mir-126-5p, hsa-mir-196b-5p, hsa-mir-584, hsa-mir-141-5p, and hsa-mir-9-3p were differentially expressed prior to correction of the data for multiple testing, suggesting that these miRNAs might be interesting targets for further evaluation in a larger sample cohort (Appendix A).

### 3.3. Differentially Expressed miRNAs in Plasma of Women with Endometriosis

We additionally analyzed hsa-mir-135a in 16 plasma samples of all endometriosis patients and 16 controls. We could show a significant increase in women with endometriosis compared to controls in plasma (Figure 2A adj. *p* = 0.048). This finding was independent of the disease stage (Figure 2B). In contrast, a significant upregulation of plasma hsa-mir-135a was only observed in the secretory phase although the proliferative phase showed a similar trend (Figure 2C).

### 3.4. Diagnostic Power of hsa-miR135a for Diagnosis of Endometriosis

To evaluate the diagnostic power of differentially expression of hsa-mir-135a, we performed receiver operating curve (ROC) analysis. In saliva, this analysis revealed that hsa-mir-135a has putative diagnostic potential (Figure 3A). The area under the curve (AUC) was 0.801 (*p*-value = 0.003) and sensitivity and specificity were 70.6% and 88.2%, respectively, with a cutoff of 0.08838 (normalized mean fluorescent intensity). We then performed the same analysis on the plasma data for hsa-mir-135a. This indicated that hsa-mir-135a also has putative diagnostic potential in plasma (Figure 3B). In this case, the AUC was 0.815 (*p*-value = 0.002) and sensitivity and specificity were 68.8% and 81.2%, respectively, with a cutoff of 0.4639 (normalized mean fluorescent intensity).

## 4. Discussion

In saliva, the high abundance of bacterial content, high enzymatic activity, and relatively low stability of salivary mRNAs create some challenges in using this biofluid as a source of diagnostic biomarkers. However, salivary miRNAs tend to be encased in exosomes, increasing their stability and attractiveness as biomarkers in this body fluid [37,38,39]. Currently, in the context of endometriosis, only one study has been published evaluating the differences in the levels of expression of salivary miRNAs and their putative application as a diagnostic biomarker for the disease [40]. The authors performed saliva miRNA sequencing and computed the saliva-based diagnostic signature for endometriosis based on a cohort of 200 patients and controls. They identified an endometriosis saliva signature of 109 miRNAs. However, these 109 miRNAs are not named, and the raw data are not available in public databases, making it difficult for independent groups to validate this data. In contrast to this high-throughput screening “omics” approach, we used a focused panel of 28 circulating miRNAs, from which 25 had been previously shown to have promising diagnostic properties for women with endometriosis. We identified hsa-mir-135a as being differentially expressed and upregulated in saliva of women with endometriosis, showing high sensitivity and specificity in distinguishing between women with and without the disease (AUC = 0.801, 70.6% sensitivity and 88.2% specificity, a cutoff of MFI = 0.08838). This increase is saliva was seen in both the proliferative and secretory phases of the menstrual cycle. In concordance with our results in saliva, we found that hsa-mir-135a is significantly upregulated in the plasma of our endometriosis patient cohort. Similar to saliva, the disease stage did not affect this difference. However, in contrast to saliva, in plasma, we found significant upregulation only in the secretory phase, although the trend was similar in the proliferative phase. Our results conflict with a previous paper that showed serum levels of hsa-mir-135a are significantly downregulated in women with endometriosis, specifically in the secretory phase [28]. In our study, we used the Bonferroni–Holm algorithm to correct for multiple testing. This previous study examined the expression of 7 different miRNAs but did not adjust for multiple testing, which would have made difference in hsa-mir-135a nonsignificant (*p* = 0.025; after Bonferroni–Holm adjustment *p* = 0.150). Therefore, although this was a valuable exploratory study, the results should be interpreted with care.

Tissue or body fluid specific miRNA expression patterns have been reported for both healthy individuals [41] and endometriosis patients [42]. There are several examples of miRNAs that show conflicting expression patterns between serum and plasma of endometriosis patients, such as hsa-mir-145 and hsa-mir-199a [25,27,29,30,43]. Other conflicting expression patterns have also been reported in other tissues and body fluids in endometriosis patients. For example, hsa-mir-106b has been reported to show increased expression in ectopic lesions [44] and peritoneal fluid of endometriosis patients [45] but decreased expression in the blood [32]. These examples could be explained by tissue- or body-fluid-specific differences, although differences in the patient cohorts used and the methodological and analytical approaches taken may also influence the outcome.

Hsa-mir-135a has been reported to show altered expression in eutopic and ectopic endometrium of endometriosis patients. In the biggest study, which included 50 controls and 32 eutopic samples, elevated levels of hsa-mir-135a in the eutopic endometrium of endometriosis patients were detected, compared to controls [6]. This was confirmed by a subsequent study, which also found that hsa-mir-135a was upregulated in eutopic endometrium [46]. While in the first study, the hsa-mir-135a expression was reported to be increased in the proliferative phase [6], in the second study, expression was upregulated in the secretory phase, although the proliferative phase showed a similar trend [46]. An independent study focusing solely on endometriosis patients found that hsa-mir-135a expression was significantly reduced in ectopic compared to eutopic endometrium [47]. Similarly, a reduction of hsa-mir-135a expression was shown in ectopic sample tissues of patients compared to controls [46]. We speculate that elevated hsa-mir-135a levels observed in patient eutopic endometrium in these studies may be connected to the increased levels of hsa-mir-135a that we see in saliva and plasma, but this would have to be experimentally shown.

Hsa-mir-135a and *HOXA10* show an inverse correlation in their expression patterns in endometriosis, with *HOXA10* being significantly decreased in the patient eutopic endometrium in the secretory phase, and significantly increased in the ectopic lesion in both the proliferative and secretory phases [46]. Furthermore, in vitro experiments have indicated that hsa-mir-135a miRNA regulates *HOXA10* transcript levels [6]. In the context of endometriosis, altered expression of *HOXA10* has been shown to be associated with reduced endometrial receptivity and ectopic lesion development [48,49], indicating a potential functional role in endometriosis for hsa-mir-135a in regulating this gene.

To date, a number of studies including our own have evaluated the potential of miRNAs from bodily fluids as diagnostic biomarkers. These studies have identified single miRNAs or panels of miRNAs that show promise as diagnostic markers for endometriosis [50]. However, although these data support the potential of miRNAs as diagnostic markers, there is currently a limited overlap between the targets identified by the different studies. This may be due to methodological differences and the heterogeneity of study design, biological samples taken, and patient cohorts.

In this study, we used FireFly technology (Abcam) to assess expression of selected miRNAs in saliva and plasma of endometriosis patients. The results in Appendix A show that in saliva, only 16 out of 34 samples have hsa-miR-135a MFI > 0 and that the overall expression is relatively low. However, 72% of the samples with MFI ≤ 0 hsa-miR-135a expression belong to the group of women without endometriosis and only 28% to women with the disease, which is unlikely to occur by chance. The FireFly technology assesses background signals using negative control wells, which are then subtracted from the signals at each target level to exclude unreliable signals. This technology is based on optical liquid stamping with a single round of amplification, allowing to measure targets close to their real abundance in the sample, unlike qPCR-based miRNAs arrays that rely on amplification. The high sensitivity of this technology, which is in the attomole range, allows lowly abundant miRNAs to be detected with high reproducibility [51].

This work revealed a putative miRNA endometriosis biomarker in saliva and plasma, body fluids that can be easily and noninvasively (saliva) or minimally invasively (plasma) collected to potentially aid in the diagnosis of endometriosis. However, there are several limitations that need to be addressed. A major limitation of this study is the low number of samples. Albeit well characterized, our control population included women with other benign gynecologic diseases, such as ovarian cysts, uterine fibroids, or unexplained infertility, which may have impacted the levels of saliva and plasma miRNAs. Thus, the results of our analysis have to be interpreted carefully with this in mind. These limitations can be overcome by validating the current data in a larger cohort, following the recommendations of the World Endometriosis Research Foundation for biomarker studies [20,52,53].

In conclusion, hsa-mir-135a shows potential as a diagnostic biomarker in saliva and plasma for endometriosis, but further studies are needed in order to validate its clinical value for this disease.

## Figures and Tables

**Figure 1 biomolecules-12-01144-f001:**
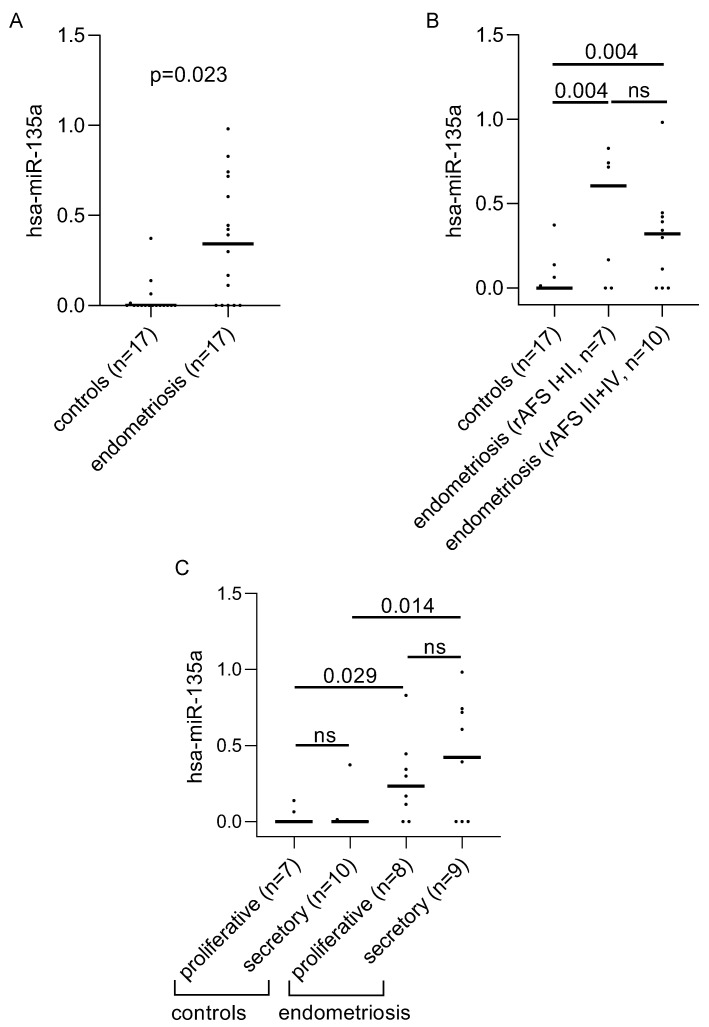
Saliva hsa-miR-135a levels are increased in women with endometriosis. (**A**) Graphical representation of the differences in the levels of expression of hsa-mir-135a in the saliva of women with and without endometriosis. (**B**) The influence of the disease stage on hsa-mir-135a expression levels. (**C**) The effect of menstrual cycle phase on hsa-mir-135a levels in saliva. Significant differences between the groups were detected by a Mann–Whitney *U*-test followed by correction for multiple testing applying Bonferroni–Holm algorithms. The adjusted *p*-values of each comparison are shown on the graphs. Abb.: ns = not significant.

**Figure 2 biomolecules-12-01144-f002:**
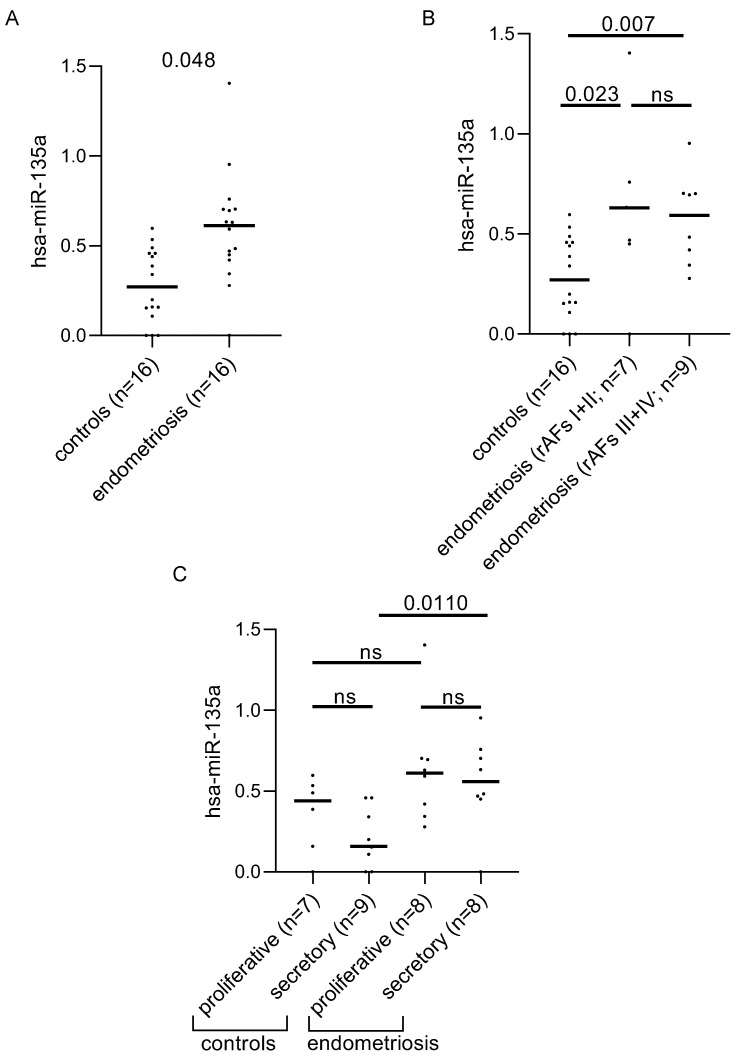
Plasma hsa-miR-135a levels are increased in women with endometriosis. (**A**) Graphical representation of the differences in the levels of expression of hsa-mir-135a in the plasma of women with and without endometriosis. (**B**) The influence of the disease stage on hsa-mir-135a expression levels. (**C**) The effect of menstrual cycle phase on hsa-mir-135a levels in plasma. Significant differences between the groups were detected by a Mann–Whitney *U*-test followed by correction for multiple testing applying Bonferroni–Holm algorithms. The adjusted *p*-values of each comparison are shown on the graphs. Abb.: ns = not significant.

**Figure 3 biomolecules-12-01144-f003:**
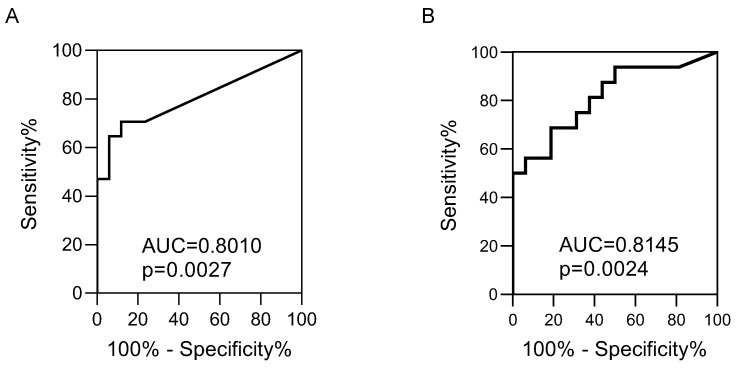
The predictive value of hsa-miR-135a for diagnosis of endometriosis expressed by ROC-curve analysis is shown. (**A**) ROC analysis for saliva. (**B**) ROC analysis for plasma. The AUC and the *p*-value are indicated on the graph.

**Table 1 biomolecules-12-01144-t001:** Overview of previously reported putative noninvasive miRNAs biomarkers for endometriosis that were included in the focused miRNA panel in this study.

miRNAs (DE in EM)^ref.^	Sample Type	Sample Size	rASRM ^1^	AUC ^2^ (Highlighted miRNAs)	Method	Normalization
**hsa-miR-16-5p, hsa-miR-191-5p, hsa-miR-195-5p**, hsa-miR-15b-5p, hsa-195-5p, hsa-362-5p:(up) [22]	plasma	53	nd	0.9	qPCR based array	hsa-miR-132
**hsa-miR-17-5p, hsa-miR-20a, hsa-miR-22-3p**:(down) [23]	plasma	40	III–IV	0.74-0.90	Microarray, qPCR	hsa-miR-16
**hsa-miR-141-3p, hsa-miR- 200a-3p**:(down) [24]	plasma	126	I–IV	0.71-0.76	qPCR	hsa-miR-30e, hsa-miR-99a
hsa-miR-145-5p (up), hsa-miR-31-5p (down) [25]	plasma	78	I–IV	nd	qPCR	hsa-miR-103-3p
**hsa-miR-143-3p, hsa-miR-20a-5p, hsa-103a-3p**: (down) [26]	plasma	106	nd	0.71-0.94	NGS, qPCR	hsa-miR-148b-3p, hsa-miR-30e-5p
**hsa-miR-154-5p, hsa-miR-378a-3p, hsa-miR-196b-5p (down), hsa-miR-33a-5p (up)** [13]	**plasma**	**92**	**I**–**IV**	**0.72**	**qPCR based array**	**hsa-miR-199a**
**hsa-miR-122**: (up) hsa-miR-9-3p, hsa-miR-141-5p, **hsa-miR-145-3p:** (down) [27]	serum	85	I–IV	0.83-0.99	Microarray, qPCR	U6snRNA
hsa-miR-135a, **let-7d-5p**: (down) [28]	serum	48	III–IV	0.91	qPCR	U6snRNA
hsa-miR-92a-3p: (up) [29]	serum	65	II–IV	nd	Array, qPCR	18sRNA
hsa-145-5p: (up) [30]	serum	48	III–IV	nd	Array, qPCR	U6snRNA
hsa-miR-15b-5p, hsa-20a-5p:(down) [31]	serum	50	I–II	nd	Solexa sequencing, qPCR	cel-miR-39
hsa-miR-93-5p, hsa-miR-126-5p:(down) [32]	blood	12	I–IV	nd	qPCR based Array	nd

EM: endometriosis. nd: not defined. DE: differentially expressed. ^1^ revised American Society of Reproductive Medicine endometriosis stage classification. ^2^ the area under the roc curve.

**Table 2 biomolecules-12-01144-t002:** Patient characteristics.

Characteristics	Controls (*n* = 17)	Endometriosis (*n* = 17)	adj. *p*-Value
Age (years ± SD)	37.4 ± 8.8	35.0 ± 7.7	0.413 *
BMI (body mass index)	25.2 ± 5.1	22.3 ± 3.5	0.062 *
Cycle phase:			
proliferative	7 (41.2%)	8 (47.1%)	0.730 ^+^
secretory	10 (58.8%)	9 (52.9%)	
rASRM ^1^ classification:			
I–II	-	7(41.2%)	
III–IV	-	10 (58.8%)	
Controls co-morbitities:			
adenomyosis	1 (5.9%)	-	
benign cysts	4 (23.5%)	-	
uterine fibroids	7 (41.2%)	-	
other	5 (29.4%)	-	
Pain Score (VAS ^2^)	4.53 ± 2.98	5.15 ± 2.36	0.508 *

^1^ revised American Society of Reproductive Medicine endometriosis stage classification. ^2.^ Visual Analogue Scale (VAS) measures pain intensity. ^+^ Pearson's chi-squared test. * *T*-test.

## Data Availability

Not applicable.

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
