# Peer review of "Hsa-mir-135a Shows Potential as A Putative Diagnostic Biomarker in Saliva and Plasma for Endometriosis"

_biomolecules, 2022, doi:10.3390/biom12081144_

Round 1

Reviewer 1 Report

The main problem of this work is that the sample of patients is small for an adequate assessment of the results obtained, given the large spread of values in the level of hsa-miR-135a expression. therefore, statements about the absence of the influence of the menstrual cycle phase and the severity of the disease on the miRNA level look unconvincing. Finally, as follows from Table S1B, hsa-miR-135 is expressed in salvia at a very low level, and may not be detected in all patients; it is obvious that such results have a high risk of accidental releases.

In order for the work to be published in a highly rated journal, it is necessary either to increase the number of samples, or to verify the results obtained, at least related to hsa-miR-135a, by another, more sensitive method

Author Response

Comments

Comment 1: The main problem of this work is that the sample of patients is small for an adequate assessment of the results obtained, given the large spread of values in the level of hsa-miR-135a expression. Therefore, statements about the absence of the influence of the menstrual cycle phase and the severity of the disease on the miRNA level look unconvincing.

Response: Our patient cohort was sufficiently large to allow us to identity hsa-miR-135a as a putative endometriosis biomarker for saliva with high sensitivity and specificity (AUC=0.801, 70.6% sensitivity and 88.2% specificity, a cut-off of MFI =0.08838). However, we agree that a larger study to validate our findings would be needed to confirm that expression of hsa-mir135a is not influenced by the menstrual cycle phase and disease severity.

Comment 2: Finally, as follows from Table S1B, hsa-miR-135 is expressed in salvia at a very low level, and may not be detected in all patients; it is obvious that such results have a high risk of accidental releases.

Response: The results in Table S2 show that only 16 out of 34 samples have hsa-miR-135a MFI>0 and that the overall expression is relatively low. However, 72% of the samples with MFI≤0 hsa-miR-135a expression belong to the group of women without endometriosis and only 28% to women with the disease, which is unlikely to be accidental. The FireFly technology is also assesses background signals using a negative control wells, which are automatically subtracted from the signals at each target level, thereby excluding unreliable signals. It is also important to note that this technology is based on optical liquid stamping that means stamping microparticle structures onto photosensitive fluids with single round of amplification, which allows to measure targets close to their real abundance in the sample and that these results can’t be compared to data obtained from qPCR based miRNAs arrays that rely on amplification. The high sensitivity of this technology, which is in the range of attomole, indicates that the low abundant miRNAs can be detected with high reproducibility (Kilic et al 2018 http://dx.doi.org/10.1016/j.bios.2017.08.007, https://docs.abcam.com/pdf/kits/cellular-mirna-assay-technology-note.pdf

Comment 3: In order for the work to be published in a highly rated journal, it is necessary either to increase the number of samples, or to verify the results obtained, at least related to hsa-miR-135a, by another, more sensitive method.

Response: The impact of this study is that it reports a putative single miRNA biomarker in saliva, a previously poorly investigated body fluid in endometriosis that is easily and safely accessible for biomarker screening. We acknowledge in the discussion that utility of hsa-miR-135a as a biomarker of endometriosis in saliva has to be validated in a carefully designed study with an independent patient cohort and sufficient sample number. However, we believe the potential impact of our finding justifies publication in Biomolecules.

Reviewer 2 Report

This study is focused on the new potential biomarker for endometriosis. In this paper, authors identified variation in the level of the microRNA termed Has-Mir-135a in saliva of patients with endometriosis – also, authors claimed that additional studies are needed to validate its clinical value as a non-invasive biomarker for this disease.

 Although it is interesting, there are some adjustments to be addressed regarding this manuscript.

There are some typed errors and that needs major improvements. A careful revision with a native speaker will benefit the entire paper. 

 All inclusion/exclusion criteria for patients were carefully followed over the study.

 For figure 1C, it would be interesting to see the statistical significance between the same menstrual phases comparing control vs. endometriosis.

 Authors stated in discussion that “The lack of correlation between the changes in the levels of expression of hsa-mir-135a in saliva and serum of women with endometriosis is currently unclear”. Although it is well recognized certain miRNAs with altered expression in blood samples, it would be interesting to reproduce the same multiplex panel using blood samples of patients, since the method and accuracy is different from PCR and microarray analysis. Also, this could reveal which are the miRNA levels with correspondence in saliva and plasma of patients with endometriosis. If this additional analysis is not possible, a better discussion could be given.

 Based on the fact that major limitation is the number of samples and associated benign disease, the information that this is a “preliminary study” should also be referred in the title of the manuscript.

Author Response

Comments

This study is focused on the new potential biomarker for endometriosis. In this paper, authors identified variation in the level of the microRNA termed Has-Mir-135a in saliva of patients with endometriosis – also, authors claimed that additional studies are needed to validate its clinical value as a non-invasive biomarker for this disease.

Although it is interesting, there are some adjustments to be addressed regarding this manuscript.

Comment 1: There are some typed errors and that needs major improvements. A careful revision with a native speaker will benefit the entire paper.

Response: A native speaker has now gone carefully through the paper and made extensive editing changes to improve the text and correct errors.

Comment 2: All inclusion/exclusion criteria for patients were carefully followed over the study. For figure 1C, it would be interesting to see the statistical significance between the same menstrual phases comparing control vs. endometriosis.

Response: We now state the level of statistical significance between controls and endometriosis with the same menstrual phase in figure 1C.

Comment 3: Authors stated in discussion that “The lack of correlation between the changes in the levels of expression of hsa-mir-135a in saliva and serum of women with endometriosis is currently unclear”. Although it is well recognized certain miRNAs with altered expression in blood samples, it would be interesting to reproduce the same multiplex panel using blood samples of patients, since the method and accuracy is different from PCR and microarray analysis. Also, this could reveal which are the miRNA levels with correspondence in saliva and plasma of patients with endometriosis. If this additional analysis is not possible, a better discussion could be given.

Response: We agree with the reviewer that it would be interested to also examine expression of hsa-mir-135a in blood or serum in order to compare to saliva, but this is outside the scope of this study. However, in response to this comment we have altered and extended the discussion to explore how tissue-specific differences, methodological differences, or differences in the patient cohort characteristics can lead to a different expression pattern for the same miRNA in different studies.

Comment 4: Based on the fact that major limitation is the number of samples and associated benign disease, the information that this is a “preliminary study” should also be referred in the title of the manuscript.

Response: We state in the title “Hsa-miR-135a is a putative non-invasive biomarker for endometriosis in saliva”. By calling hsa-miR-135a a putative biomarker we clearly indicate that more work is required to confirm this. Furthermore, in the discussion we acknowledge the weaknesses of our study, principally the relatively small sample size, and recommend the work be validated in an independent study.

Reviewer 3 Report

In this paper the authors presented results showing that hsa-mir135 was upregulated in saliva of women with endometriosis suggesting it can be potential non-invasive biomarker to detect this gynecological disease.  

Some minor comments:

Discussion was overall brief, the author should elaborate why the sensitivity and specificity levels  differ in saliva compare to serum levels. Also discuss the importance of conducting an optimized experimental method for confirming these findings in an independent population.

-

Author Response

Comments

In this paper the authors presented results showing that hsa-mir135 was upregulated in saliva of women with endometriosis suggesting it can be potential non-invasive biomarker to detect this gynecological disease.

Some minor comments:

Comment 1: Discussion was overall brief, the author should elaborate why the sensitivity and specificity levels differ in saliva compare to serum levels.

Response: We have extended this section in the discussion to better explain the nature of the discrepancies in the levels of expression of hsa-miR-135a in saliva and serum. In this study we showed that hsa-miR-135a was upregulated in the saliva of endometriosis patients, and we performed receiver operating characteristics (ROC) analysis to determine the sensitivity and specificity of its diagnostic value for endometriosis. This was not done in the study by Cho et al 2015 that showed hsa- miR-135a expression in down-regulated in serum, so we cannot compare the sensitivity and specificity of hsa-miR-135a diagnostic value in saliva and serum.

Comment 2: Also discuss the importance of conducting an optimized experimental method for confirming these findings in an independent population.

Response: We have now modified and extended the discussion including to discuss the importance of carefully designed validation studies to test biomarker candidates in an independent patient cohort.

Round 2

Reviewer 1 Report

The main and unremovable problem of the work is the quality of the obtained data.

      First, there are some issues with normalization. Spike-in control can be used, and such works are published, but the disadvantages of this approach are also obvious. The manufacturer's website (https://www.abcam.com/kits/fireplex-mirna-assay#2) contains the following information: “If the synthetic target of choice is included within your custom or focused panel, the assay can detect that target and use it for normalization. However, in our experience normalizing using the geometric average of all detected targets is generally preferable, since synthetic spike-ins may partition within the sample in ways that are distinct from the endogenous miRNAs.” In the current version of the article, the authors additionally use the arithmetic mean of the expression values ​​obtained for all miRNAs (Supplementary Figure 1) for normalization, however, it seems it is incorrect to use this parameter due to the fact that the range of values ​​is very large (from 0.028 to 1202, 317, table S1B); if normalization using the geometric mean yields a meaningful result, this objection can be lifted.

    Secondly, some doubts about the obtained results are due to the fact that most of the differences (lines 435-437) were obtained for miRNA (hsa-mir-135a, hsa-mir-126-5p, hsa-mir-196b-5p, hsa- mir-584and hsa-mir-9-3p), the expression of which is determined in no more than 50% of the samples (Supplementary Table S2) and/or at a very low level (0.106-0.424 MFI) (Supplementary Table S1B). Therefore, in our opinion, such results require mandatory verification by an alternative method (or, perhaps, the authors should have used the FirePlex miRNA Assay for purified RNA (the manufacturer's website states that «If you have 100 pg–500 ng (in a 25 μL volume ) we recommend use of the FirePlex miRNA Assay for purified RNA»). If the authors can prove the reliability of their data in another way, this objection can also be removed.

     Sorry for unconvinience, but as presented now, the results can only be considered as preliminary.  

In section 2.3 it is worth adding information about how the plasma was analyzed (probably authors also used 30 µl of prepared plasma, but still it is worth mentioning.

Author Response

Reviewer 1 comments

The main and unremovable problem of the work is the quality of the obtained data.

Comment 1

      First, there are some issues with normalization. Spike-in control can be used, and such works are published, but the disadvantages of this approach are also obvious. The manufacturer's website (https://www.abcam.com/kits/fireplex-mirna-assay#2) contains the following information: “If the synthetic target of choice is included within your custom or focused panel, the assay can detect that target and use it for normalization. However, in our experience normalizing using the geometric -average of all detected targets is generally preferable, since synthetic spike-ins may partition within the sample in ways that are distinct from the endogenous miRNAs.” In the current version of the article, the authors additionally use the arithmetic mean of the expression values ​​obtained for all miRNAs (Supplementary Figure 1) for normalization, however, it seems it is incorrect to use this parameter due to the fact that the range of values ​​is very large (from 0.028 to 1202, 317, table S1B); if normalization using the geometric mean yields a meaningful result, this objection can be lifted.

Response:

As the reviewer acknowledges, the spike-in control can be used for normalization of miRNA expression data, and in our case, we believe it is the most appropriate normalization control. Initially we did not use the geometric mean for normalization as our data contains zero values, meaning the standard geometric mean will be zero and therefore it cannot be used for normalization. As an alternative, we normalized to the arithmetic mean, which confirmed a significant increase in hsa-miR-135a expression in the saliva of endometriosis patients. However, we acknowledge the reviewers concerns about using the arithmetic mean for normalization, given the large range of expression values. Therefore, we also normalized to a non-standard geometric mean calculated following the approach of Habib (https://www.arpapress.com›IJRRAS_11_3_08) where zero values are removed. This also indicated that hsa-miR-135a expression is increased in the saliva of endometriosis patients. We now include the re-analyzed data normalized to both the arithmetic mean and the non-standard geometric mean in Supplementary Figure 1, and have modified the methods and results sections of the manuscript accordingly (lines 146-152 and lines 179-182).

Comment 2

    Secondly, some doubts about the obtained results are due to the fact that most of the differences (lines 435-437) were obtained for miRNA (hsa-mir-135a, hsa-mir-126-5p, hsa-mir-196b-5p, hsa- mir-584and hsa-mir-9-3p), the expression of which is determined in no more than 50% of the samples (Supplementary Table S2) and/or at a very low level (0.106-0.424 MFI) (Supplementary Table S1B). Therefore, in our opinion, such results require mandatory verification by an alternative method (or, perhaps, the authors should have used the FirePlex miRNA Assay for purified RNA (the manufacturer's website states that «If you have 100 pg–500 ng (in a 25 μL volume ) we recommend use of the FirePlex miRNA Assay for purified RNA»). If the authors can prove the reliability of their data in another way, this objection can also be removed.

     Sorry for unconvinience, but as presented now, the results can only be considered as preliminary. 

Response:

We appreciate the reviewers concern on the need for validation of our data. However, we believe in the latest version of the paper we provide this validation in the form of FirePlex miRNA expression data from the plasma of endometriosis patients and controls. We show that hsa-miR-135a expression is increased in the plasma of endometriosis patients, supporting our finding in saliva (Figure 2). Furthermore, with ROC analysis we show that increased hsa-miR-135a expression in plasma also has predictive value for diagnosis of endometriosis (Figure 3B). This data both validates the saliva finding and extends the value of our study showing that hsa-miR-135a is a putative biomarker for endometriosis in both saliva and plasma.

Reviewer 2 Report

No additional comments

Author Response

We thank the reviewer their effort in reviewing out paper